# CoTabBench: A Real-World Complex Table Benchmark for Question Answering over Weakly-Structured and Heterogeneous Tables

## Abstract

Recent advancements in Large Language Models (LLMs) have significantly propelled their capabilities in table-based question answering. However, existing benchmarks predominantly feature well-structured tables, failing to address the complexities of real-world data, which is often weakly-structured and contains highly heterogeneous content. This discrepancy limits the evaluation of model robustness on diverse and challenging formats, such as tables with intricate layouts and varied data types found in scientific papers or financial reports. To bridge this gap, we introduce **CoTabBench**, a large-scale, multi-domain, and intricate benchmark featuring over 2,700 real-world, weakly-structured tables and more than 8,600 question-answer pairs spanning 10 distinct domains. We further propose a novel complexity assessment framework, which quantitatively validates the inherent structural and content-based challenges within CoTabBench. Furthermore, we introduce **CoTabInstruct**, a large-scale training corpus with over 11,000 tables, and present **CoTabLLM**, a 7B model trained on it that outperforms even leading models like GPT-4.1 on our benchmark. Extensive experiments reveal a significant performance degradation for state-of-the-art models on CoTabBench, highlighting its critical role in advancing robust, real-world table understanding.

## 1 Introduction

Recent advancements in Large Language Models (LLMs) have significantly propelled the field of Table Question Answering (TableQA). A series of established benchmarks, such as TableBench, DocTabQA, and SpreadsheetBench (Wu et al., 2025; Tang et al., 2024; Wang et al., 2024a; Ma et al., 2024; Herzig et al., 2020), have demonstrated the formidable capabilities of these models in processing structured tabular data and executing complex reasoning. Research in this domain originated from foundational explorations into parsing semi-structured web tables (Pasupat & Liang, 2015b; Herzig et al., 2020) and has progressively expanded to encompass more challenging tasks, including fact verification, hybrid reasoning, and domains requiring sophisticated numerical computation (Chen et al., 2020; 2021b; Zhu et al., 2021; Lu et al., 2023; 2025).

However, despite the escalating complexity of these benchmarks across various dimensions, the underlying tabular data they employ remains predominantly regular and clean. This creates a significant disconnect from the tables commonly encountered in the real world, which are often characterized by more intricate, arbitrary structures and heterogeneous content. This discrepancy raises a critical research question: How robust are state-of-the-art models when confronted with the structural and content complexities inherent in tables sourced from scientific papers, government reports, and other real-world documents?

While existing research has acknowledged parts of this problem, efforts to address its full scope have been fragmented. On one hand, some studies have focused on specific structural difficulties, such as the hierarchical layouts common in statistical reportsLiu et al. (2023); Holecek et al. (2019); Chen et al. (2021c); Zhu et al. (2021); He et al. (2025). On the other hand, different research has concentrated on domain-specific knowledge barriers, constructing datasets for specialized fields like

| Dataset | Test Set Samples | Unique Tables | Avg. Tokens | Complexity | | Source |
|---|---|---|---|---|---|---|
| | | | | Structure | Content | |
| WTQ | 4.3K | 0.4K | 1600.5 | Regular | T, N | Wikipedia |
| TableFact | 6.8K | 1.7K | 387.9 | Regular | T, L, N | Wikipedia |
| FinQA | 1.1K | 0.3K | 114.2 | Regular | T, C, F | Financial |
| FeTaQA | 2.0K | 1.9K | 417.2 | Regular | T, L, N | Wikipedia |
| AIT-QA | 0.5K | 0.1K | 214.4 | Hierarchical | T, C | Airlines |
| TableBench | 0.8K | 0.5K | 537.3 | Regular | T, L, N | Aggregated |
| HiTab | 1.5K | 0.5K | 654.0 | Hierarchical | T, L, C | Wikipedia |
| TABMWP | 7.6K | 7.5K | 58.4 | Regular | N, F | Math Exam |
| TAT-QA | 1.6K | 0.2K | 209.5 | Regular | T, C | Financial |
| InfoTabs | 5.4K | 0.6K | 296.1 | Regular | T, L, N | Wikipedia |
| PubHealthTab | 1.9K | 0.2K | 790.9 | Regular | T, L, C, S | Public Health |
| **CoTabBench** | **8.6K** | **2.7K** | **952.3** | **Irregular** | **T,L,N,C,F,S** | **ArXiv and Web** |

Table 1: Comparison with existing datasets. We define content types with symbols: **T** (Text), **L** (Long-form text), **N** (Numeric), **C** (Complex Numeric), **F** (Formulas), and **S** (Special Symbols).Structure denotes the primary table structure targeted by each dataset.

finance or aviationKatsis et al. (2022); Akhtar et al. (2022); Bai et al. (2024). Although these studies highlight important dimensions of the table understanding challenge, they fail to capture a more pervasive and chaotic reality: real-world tables often simultaneously exhibit both weakly-structured characteristics—such as irregular layouts, merged cells, and multi-row entries—and content heterogeneity, where cells contain a complex mixture of long-form text, mathematical formulas, and special symbols, as illustrated in Figure 5. Consequently, a comprehensive benchmark that systematically evaluates model performance against this compound, real-world challenge remains a critical gap in the field.

To bridge this gap, we introduce **CoTabBench**, a large-scale, multi-domain question-answering benchmark specifically designed for weakly-structured and heterogeneous tables. Sourced directly from 10 authentic academic and applied domains, CoTabBench comprises over 2,700 free-form tables and more than 8,600 question-answer pairs. To systematically quantify the challenges posed by these tables, we developed a novel **Complexity Assessment Framework**. This framework provides a high-level evaluation of a table's intrinsic difficulty along the two core dimensions of structural irregularity and content heterogeneity, assessing inherent challenges in layout, hierarchy, and content diversity. This framework not only theoretically validates the challenging nature of CoTabBench but also offers a standardized methodology for future table complexity research.

Subsequently, we created a companion large-scale training set containing over 11,000 tables and 32,000 examples. Leveraging this resource, we developed **CoTabLLM**, a 7B-parameter baseline model that outperforms leading models such as GPT-4.1 on our benchmark. Extensive experiments involving more than 20 state-of-the-art LLMs demonstrate a significant performance decline on CoTabBench, thereby emphasizing the necessity of our proposed benchmark for advancing robust, real-world table understanding.

We summarize our contributions as follows:

- We propose CoTabBench, a large-scale, multi-domain benchmark for weakly-structured tables, and propose a novel Complexity Assessment Framework to quantitatively validate its inherent structural and content-based challenges.

- We introduce CoTabInstruct, a comprehensive instruction training corpus comprising over 11,000 tables and 32,000 QA pairs. We also present CoTabLLM, a 7B parameter model trained on CoTabInstruct, which serves as a robust baseline.

- We systematically evaluate over 20 leading LLMs on CoTabBench, revealing significant performance gaps. Our extensive experiments establish CoTabBench as a critical testbed for measuring the gap between current models and the demands of real-world table understanding.

## 2 COTABBENCH CONSTRUCTION

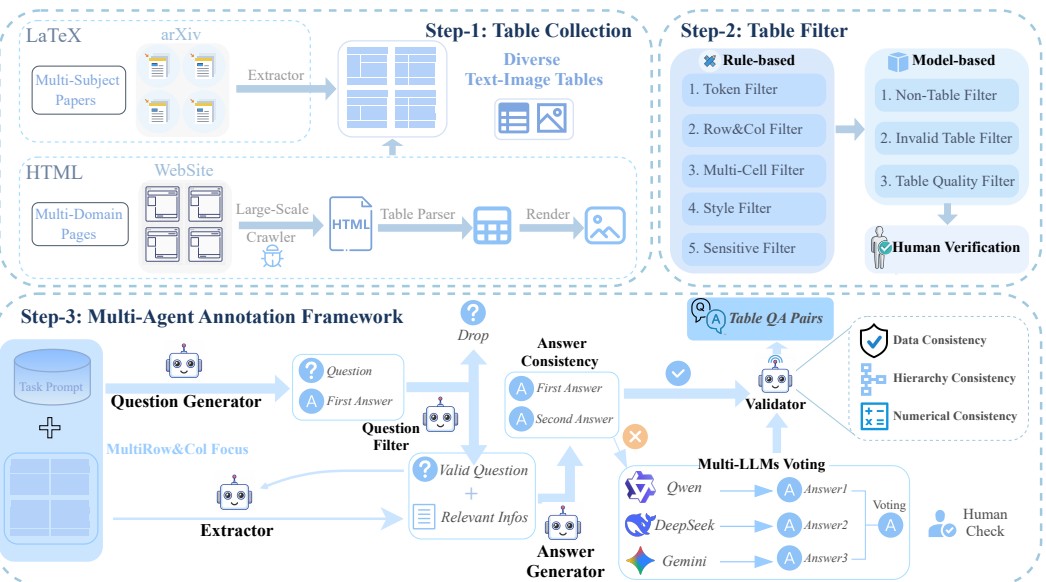

Figure 1: A comprehensive overview of the annotation framework.

To systematically address the prevalent yet under-explored challenge of question answering over complex real-world tables, we propose a rigorous, multi-stage construction pipeline. Our process begins by collecting tables in the wild from scientific literature and public websites, followed by rigorous filtering to eliminate malformed layouts and overly simplistic tables. For these curated tables, we define three core question-answering tasks with specifications tailored to address their inherent challenges. Finally, we employ a novel Multi-Agent Annotation Framework (Figure 1) to generate high-quality QA pairs, yielding our two primary contributions: the benchmark dataset comprising over 2,700 tables and 8,600 QA pairs, and the instruction set comprising over 11,000 tables and 32,000 QA pairs.

### 2.1 COMPLEX TABLE COLLECTION

Departing from prior work that relies on existing or synthetic datasets, we source tables directly *in the wild* using a dual-pronged strategy to ensure authentic complexity. First, we extract tables from their native LaTex source within the **DocGenome** (Xia et al., 2024) dataset, a large-scale corpus of parsed arXiv documents, preserving the complex structural semantics (e.g., multicolumn and multirow) often lost in PDF conversion. Second, we deploy a large-scale crawler to gather HTML tables from public websites in domains like government, finance, and medicine, which contribute significant content heterogeneity. This dual-source approach provides a challenging and representative foundation for our benchmark, capturing both the intricate layouts of scientific literature and the diverse content of real-world web data.

### 2.2 TABLE FILTERING

To ensure only high-quality and genuinely complex tables form our dataset, the raw collection is pruned by a rigorous three-stage filtering pipeline. The process begins with rule-based heuristics to discard tables with insufficient content or clear structural anomalies, followed by a dual model-based assessment where an LLM evaluates semantic coherence and an MLLM validates the visual table. Finally, a crucial human verification stage serves as the ultimate quality gate to remove any subtle, low-quality samples that eluded the automated filters.

## 2.3 TASK DESIGN

To isolate the challenges posed by complex table structures and heterogeneous content, we simplify the question-answering tasks. Instead of intricate reasoning chains, we adapt three fundamental tasks to directly probe a model's ability to overcome the intrinsic difficulties of weakly-structured tables. This approach shifts the locus of difficulty from the reasoning process to the essential step of understanding the table itself, allowing us to precisely measure the impact of structural complexity on core model capabilities.

**Row and Column Counting** This task tests structural comprehension. In weakly-structured tables, a definitive physical count of rows or columns often does not exist or is misleading. The model must therefore pierce through visual artifacts like merged cells to identify the number of *logical* entities. This task directly measures a model's ability to deconstruct a table's visual representation and grasp its underlying logical schema.

**TableQA** This task requires the model to answer a natural language question based on the given weakly-structured table. The questions are designed to necessitate multi-step reasoning and precise localization, often involving querying information under hierarchical headers or within cells spanning multiple rows/columns. It fundamentally evaluates the model's ability to semantically align the question with the table's complex structure and navigate its layout to retrieve the correct answer, serving as a direct measure of comprehensive table understanding.

**Hallucination Detection** This task tests factual grounding by presenting two types of negative questions. The first, Data Absence, queries for information not present in the table. The second, a more subtle Attribution Error, asks about a value that exists in the table but proposes an incorrect association. This directly probes a model's ability to resist making false inferences induced by misleading layouts.

## 2.4 MULTI-AGENT QA ANNOTATION FRAMEWORK

Manual annotation of high-quality QA pairs incurs prohibitive costs, while naive single-LLM generation yields simplistic or factually inaccurate outputs. To bridge this quality-scale gap, we propose a Multi-Agent Annotation Framework that orchestrates a cascade of specialized LLM agents to generate, refine, and validate QA pairs for complex TableQA and Hallucination Detection tasks. Statistical counting tasks are addressed via bespoke matching rules. As shown in Figure 1, the workflow proceeds as follows:

**Generation and Filtering** Our QA generation process integrates generation with immediate self-verification to ensure high quality. First, a generator agent creates candidate question-answer pairs by targeting challenging table regions, such as multi-span cells. The question undergoes rigorous filtering to eliminate invalid queries, and for those that pass, grounded evidence is extracted from the table. Subsequently, a separate Answer Generator synthesizes a refined answer using only this extracted evidence. A critical consistency check then compares this refined answer against the initial one. Agreement between the two serves as an internal validation, while any discrepancy automatically escalates the pair to our Multi-LLM Voting module for a final, definitive decision.

**Multi-LLM Voting for Discrepancies** Discrepant pairs are escalated to a Multi-LLM Voting module, where a majority vote among answers from three independent LLMs determines the final output, enhancing reliability in ambiguous cases.

**Multi-Dimensional Automated Validation** Whether a QA pair passes self-verification or the voting stage, it must undergo a final, stringent validation by our automated **Validator** module. This module assesses the pair against three critical dimensions of consistency:

- **Data Consistency**: We programmatically verify that all factual claims in the answer exist as substrings within the table's content, ensuring the answer is directly grounded and not fabricated.

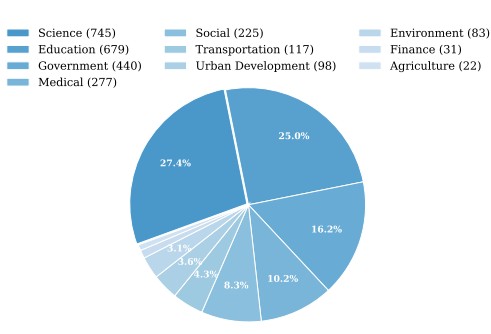

Figure 2: Domain distribution of the test tables. Science and Education constitute the largest shares, followed by Government and Medical. The remaining domains each contribute a relatively small proportion.

| Properties | Value |
|---|---|
| *Basic Statistics* | |
| # Train Set Unique Tables | 11096 |
| # Train Set Examples | 32645 |
| # Test Set Unique Tables | 2717 |
| # Test Set Examples | 8609 |
| Avg. Rows / Table | 14.9 |
| Avg. Columns / Table | 8.4 |
| Avg. Tokens / Table | 952.3 |
| # Domains | 10 |
| *Table Complexity* | |
| Avg. Multi-Row | 4.3 |
| Avg. Multi-Col | 5.1 |
| Avg. Multi-Cells | 70.6 |
| Avg. Cell Num | 89.8 |
| Ratio of Multi-Cells | 98.3% |
| *Question Categories* | |
| Row&Col Counting | 31.6% |
| TableQA | 37.7% |
| Hallucination Detection | 30.7% |

Table 2: Detailed statistics of CoTabBench

- **Hierarchical Consistency**: A rigorous bidirectional check is performed to confirm that the answer's content resides at the precise row/column coordinates specified in the question, preventing attribution errors.

- **Numerical Consistency**: For questions involving computation, the validator dynamically generates and executes Python code with 'assert' statements to verify the numerical accuracy of the answer.

**Human-in-the-Loop Final Check**  All samples in our benchmark ultimately underwent manual verification to ensure the accuracy and reliability of the evaluation set.

## 2.5 BENCHMARK AND TRAINING SET

Our multi-stage pipeline culminates in the **CoTabBench Suite**, which consists of two tightly integrated resources designed to assess and advance table QA capabilities:

- **CoTabBench**: The core evaluation benchmark, containing **2,700 tables and 8,600 QA pairs**. It is meticulously curated to serve as a rigorous testbed for measuring model performance on complex, real-world tables.

- **CoTabInstruct**: A large-scale instruction-tuning dataset with over **11k tables and 32k QA pairs**. It is released to enable the community to fine-tune models specifically for mastering the structural and content challenges prevalent in CoTabBench.

## 3 COMPLEXITY QUANTIFICATION

To empirically validate that CoTabBench is more challenging than existing benchmarks, we introduce a new quantitative framework that moves beyond anecdotal evidence. This framework systematically measures a table's intrinsic difficulty along two orthogonal dimensions: **Structural Irregularity** and **Content Heterogeneity**. This approach provides a principled basis for comparing CoTabBench against its predecessors and offers a standardized methodology for future research on complex table understanding.

### 3.1 COMPLEXITY DIMENSIONS

We define two core dimensions that capture the multifaceted challenges of real-world tables.

| Metric | WTQ | InfoTabs | CoTabBench |
|---|---|---|---|
| Avg. MultiRows | 1.36 | 0.00 | 4.33 |
| Avg. MultiCols | 0.17 | 0.20 | 5.14 |
| Avg. MultiRowCells | 4.17 | 0.00 | 13.58 |
| Avg. MultiColCells | 1.18 | 0.41 | 16.36 |
| Avg. MergedCells | 7.19 | 2.12 | 19.17 |
| Avg. Multi-Cells | 5.35 | 0.41 | 29.94 |
| MultiRow Ratio | 17.58% | 0.00% | 82.19% |
| MultiCol Ratio | 9.74% | 14.00% | 84.26% |
| MultiCell Ratio | 25.18% | 14.00% | **98.34**% |

(a) Comparison by structural statistics.

| Dimension | Metric | WTQ | InfoTabs | CoTabBench |
|---|---|---|---|---|
| Structural Irregularity | MCR | 0.07 | 0.12 | **0.31** |
| | MCS | 0.15 | 0.03 | **1.29** |
| | HDI | 2.32 | 2.02 | **3.91** |
| | Total | 2.54 | 2.17 | **5.51** |
| Content Heterogeneity | CCC | 2.81 | 2.92 | **3.00** |
| | DTD | 2.13 | 2.46 | **2.52** |
| | SDC | 3.29 | 2.96 | **4.47** |
| | Total | 8.23 | 8.34 | **9.99** |

(b) Comparison by irregularity and heterogeneity metrics.

Table 3: Overall dataset comparison of structural complexity and irregularity metrics

**Structural Irregularity** This dimension quantifies the structural irregularity of a table, specifically its deviation from a canonical two-dimensional grid. Such irregularity invalidates simple coordinate-based indexing, compelling a model to achieve a deeper semantic comprehension of the layout. It manifests in several forms: **merged cells**, which introduce complex, multi-level dependencies; **nested headers**, which necessitate hierarchical reasoning to correctly associate data cells with their complete contextual headers; and **multi-line entries**, which embed structured information within a single cell. Furthermore, this category encompasses atypical layouts that rely on visual cues for interpretation, as well as instances of **implicit or missing headers**, where cell semantics must be inferred entirely from the surrounding context. These factors collectively pose a significant challenge to a model's global structural understanding.

**Content Heterogeneity** This dimension addresses the complexity of information in both form and substance, moving beyond simple atomic values. At the micro-level, this is evident in cells containing not just atomic values but also **long-form text**, which requires summarization or extraction; **complex numerics** presented with units, ranges, or appended symbols; and specialized semantic units like **mathematical formulas** or **domain-specific symbols**, which require specialized parsing and knowledge. This often results in mixed content types coexisting within a single cell. At the macro-level, the domain-level heterogeneity of CoTabBench arises from its sourcing across 10 distinct professional fields, demanding that models generalize their understanding across diverse terminologies, conventions, and knowledge contexts.

## 3.2 COMPLEXITY ASSESSMENT FRAMEWORK

**Quantifying Structural Irregularity** Structural irregularity is the defining characteristic of **CoTabBench**. This is quantitatively evident in Table 3a: 98.34% of our tables feature merged cells, a stark contrast to just 25.18% in prior benchmarks like WTQ. To move beyond this simple prevalence and formalize the degree and nature of this complexity, we introduce the following metrics.

- **Merged Cell Ratio (MCR):** This metric measures the prevalence of merged cells, which are primary indicators of a non-grid layout. It is defined as follows:

$$\text{MCR} = \frac{\text{Number of Merged Cells}}{\text{Total Number of Cells}} \tag{1}$$

- **Merged Cell Span (MCS):** While MCR captures the frequency of merged cells, MCS quantifies their average magnitude or scale. For a table $T$ and a cell $c$, it is calculated as:

$$\text{MCS} = \frac{\sum_{c \in T}(\text{multirow}(c) - 1) + \sum_{c \in T}(\text{multicol}(c) - 1)}{\text{row\_count}(T) + \text{col\_count}(T)} \tag{2}$$

- **Header Depth Index (HDI):** This metric assesses the hierarchical complexity of the table's headers. The index is the sum of the maximum level depths for row and column headers:

$$\text{HDI} = \text{depth}(\text{row\_headers}) + \text{depth}(\text{col\_headers}) \tag{3}$$

**Quantifying Content Heterogeneity**   The complexity of cell content is semantic and nuanced, making it ill-suited for simple rule-based measurement. We therefore employ an LLM-as-a-judge methodology (Zheng et al., 2023; Chen et al., 2024; Li et al., 2024a; Gu et al., 2025), leveraging Qwen-Plus-Latest to score content along three distinct axes on a scale of 1 to 5.

- **Cell Content Complexity (CCC):** Evaluates the complexity within individual cells, ranging from simple atomic values to composite entries like long paragraphs or mathematical formulas.
- **Data Type Diversity (DTD):** Assesses the variety of data types across the table. A high score indicates a rich mixture of content, such as symbolic and formula.
- **Semantic & Domain Complexity (SDC):** Captures the reliance on domain-specific knowledge. A high score signifies a knowledge barrier requiring understanding of specialized terminology.

### 3.3   COMPLEXITY VALIDATION

Our analysis, as shown in Table 3b, demonstrates that CoTabBench sets a significantly more rigorous standard than prior benchmarks like WTQ and InfoTabs. Its **Structural Irregularity** score of **5.51** over doubles that of peers, indicating more complex layouts. For **Content Heterogeneity**, CoTabBench's top score of **9.99** reflects its diverse, knowledge-intensive sourcing from 10 domains. These results position CoTabBench as a paradigm shift, essential for evaluating models on real-world table complexity.

Furthermore, the reliability of our content-complexity measurements is supported by the strong consistency between LLM-as-a-judge and human experts: Table 5 reports per-metric Pearson correlations above 0.98 and an overall correlation of 0.973, validating the automatic scorer as a faithful surrogate for expert assessment on complex tables. Complementing this, the data-scaling curves in Figure 4 show that model performance improves monotonically as a larger proportion of CoTabInstruct is used for SFT, with gains persisting on the hard (high-complexity) subset, further confirming the effectiveness of our dataset for complex table understanding.

## 4   EXPERIMENTS

### 4.1   SETUP

To ensure a fair and standardized evaluation, we employ uniform prompt templates leveraging JSON Schema to enforce a strict, structured output format, from which the final answer is programmatically parsed. We query proprietary and open-source models via their official APIs, while specialized fine-tuned models are deployed locally on four 4x4090 GPUs using the vLLM framework for inference. Our fine-tuned model, **CoTabLLM**, was developed by applying Supervised Fine-Tuning to Qwen2.5-7B-Instruct and LLaMA3-8B-Instruct on the **CoTabInstruct** dataset with Low-Rank Adaptation (LoRA). The training was configured with a batch size of 16 and a maximum sequence length of 4096 tokens.

### 4.2   EVALUATED LLMS

We evaluate large language models covering diverse architectures and sources, with parameter sizes ranging from 7B to 235B. Our assessment includes leading proprietary (commercial) models, such as GPT-4.1, GPT-4.1-mini (OpenAI et al., 2024c;a), the Qwen-Plus/Turbo series (Yang et al., 2025), and Gemini 2.5 Flash (Comanici et al., 2025), as well as a broad set of open-source counterparts, including the Llama3 and Llama 4 series (Touvron et al., 2023; Grattafiori et al., 2024), various Qwen2.5/Qwen3 versions (Qwen et al., 2025; Hui et al., 2024; Yang et al., 2025), DeepSeek-V3 (DeepSeek-AI et al., 2025b), and Kimi K2 (Team et al., 2025). To investigate the performance of Thinking LLMs with complex reasoning capabilities, we specifically evaluate o4-mini (OpenAI et al., 2024b) and DeepSeek-R1 (DeepSeek-AI et al., 2025a), and analyze the performance of Qwen series, and Gemini 2.5 Flash with their reasoning modes enabled. Finally, we benchmark against existing table-specialized models (Su et al., 2024; Wu et al., 2025) and introduce our own fine-tuned model, CoTabLLM, to establish a robust baseline.

| Models | Counting | | | TableQA | | | Hall.Detect | | | Overall | | |
|---|---|---|---|---|---|---|---|---|---|---|---|---|
| | easy | hard | avg | easy | hard | avg | easy | hard | avg | easy | hard | avg |
| *Close-source LLMs* | | | | | | | | | | | | |
| Ⓤ GPT-4.1 | 51.1 | 51.0 | 51.1 | 54.7 | 41.4 | 47.6 | 65.4 | 57.8 | 61.5 | 57.1 | 50.1 | 53.4 |
| Ⓤ GPT-4.1-mini | 42.8 | 38.8 | 40.7 | 48.8 | 35.3 | 41.7 | 58.8 | 51.2 | 54.8 | 50.1 | 41.8 | 45.7 |
| Qwen-Turbo-Latest | 49.7 | 50.2 | 50.0 | 39.8 | 30.8 | 35.0 | 46.2 | 38.3 | 42.1 | 45.2 | 39.8 | 42.4 |
| Qwen-Plus-Latest | 54.0 | 50.4 | 52.1 | 55.5 | 41.5 | 48.1 | 68.3 | 62.5 | 65.3 | 59.3 | 51.5 | 55.2 |
| ◆ Gemini 2.5 Flash | 54.5 | 55.2 | 54.9 | 55.8 | 44.2 | 49.6 | 55.1 | 47.5 | 51.1 | 55.1 | 48.9 | 51.9 |
| *Open-source LLMs* | | | | | | | | | | | | |
| DeepSeek-V3 | 52.1 | 51.0 | 51.5 | 58.0 | 43.9 | 50.5 | 65.4 | 57.0 | 61.0 | 58.5 | 50.6 | 54.4 |
| Kimi K2 | 46.4 | 42.0 | 44.1 | 62.1 | 53.8 | 57.7 | 52.9 | 42.8 | 47.7 | 53.8 | 46.2 | 49.8 |
| Qwen2.5-7B | 32.3 | 32.8 | 32.6 | 25.6 | 18.9 | 22.2 | 42.8 | 30.2 | 36.2 | 33.6 | 27.3 | 30.3 |
| Qwen2.5-72B | 53.1 | 53.8 | 53.5 | 54.5 | 41.4 | 47.5 | 58.7 | 51.7 | 55.1 | 55.5 | 49.0 | 52.0 |
| Qwen3-8B | 46.6 | 42.6 | 44.5 | 37.8 | 30.3 | 33.8 | 34.2 | 26.3 | 30.1 | 39.6 | 33.1 | 36.1 |
| Qwen3-235B-A22B | 53.3 | 49.9 | 51.5 | 54.9 | 42.8 | 48.4 | 49.4 | 43.5 | 46.3 | 52.5 | 45.4 | 48.8 |
| ∞ Llama 3.1-8B | 8.9 | 9.1 | 9.0 | 32.2 | 21.9 | 26.8 | 15.2 | 11.8 | 13.4 | 18.8 | 14.2 | 16.4 |
| ∞ Llama 3.3-70B | 41.9 | 36.5 | 39.0 | 53.9 | 43.1 | 48.2 | 68.2 | 62.0 | 65.0 | 54.7 | 47.2 | 50.7 |
| ∞ Llama 4 Maverick | 38.1 | 50.3 | 44.6 | 65.0 | 54.5 | 59.5 | 48.5 | 37.7 | 42.9 | 50.5 | 47.5 | 49.0 |
| *Thinking LLMs* | | | | | | | | | | | | |
| Ⓤ o4-mini | 65.0 | 67.8 | 66.5 | 66.4 | 52.2 | 58.9 | 68.2 | 62.0 | 65.0 | 66.6 | 60.6 | 63.4 |
| DeepSeek-R1 | 83.8 | 85.3 | **84.6** | 64.8 | 52.4 | 58.2 | 68.3 | 62.0 | 65.0 | 72.3 | 66.6 | **69.3** |
| Qwen-Turbo-Thinking | 79.3 | 76.2 | 77.6 | 62.5 | 53.3 | 57.7 | 67.6 | 60.5 | 63.9 | 69.8 | 63.3 | 66.4 |
| Qwen-Plus-Thinking | 45.6 | 50.2 | 48.1 | 70.8 | 64.2 | **67.3** | 67.4 | 60.2 | 63.7 | 61.3 | 58.2 | 59.7 |
| *Open-source FineTuned LLMs(Non-Thinking)* | | | | | | | | | | | | |
| TableLLM-Qwen2-7B | 7.8 | 8.9 | 8.4 | 28.4 | 20.9 | 24.3 | 0.2 | 0.1 | 0.1 | 12.1 | 10.0 | 10.9 |
| TableGPT2-7B | 41.9 | 41.0 | 41.4 | 35.1 | 23.9 | 29.2 | 29.1 | 20.4 | 24.6 | 35.4 | 28.4 | 31.7 |
| **CoTabLLM-7B** | 57.8 | 56.5 | 57.1 | 37.5 | 34.8 | 36.1 | 82.4 | 75.1 | **78.6** | 59.2 | 55.5 | 57.3 |
| **CoTabLLM-8B** | 57.0 | 57.6 | 57.3 | 35.1 | 29.6 | 32.2 | 68.3 | 62.0 | 65.0 | 53.5 | 49.7 | 51.5 |

Table 4: Performance comparison of LLMs on CoTabBench. Counting: Row and Column Counting, Hall.Detect: Hallucination Detection. CoTabLLM-7B based on Qwen2.5-7B, CoTabLLM-8B based on LLaMA3-8B.

## 4.3 EVALUATION METRICS

To assess model performance on CoTabBench, we employ strict, task-specific metrics. For both the Row and Column Counting and Hallucination Detection tasks, which yield definitive ground-truth answers, we use **Accuracy**. For the TableQA task, we use **Exact Match**, calculated after applying text normalization (removing special symbols and whitespace) to both the predicted and ground-truth strings.

## 4.4 MAIN RESULTS

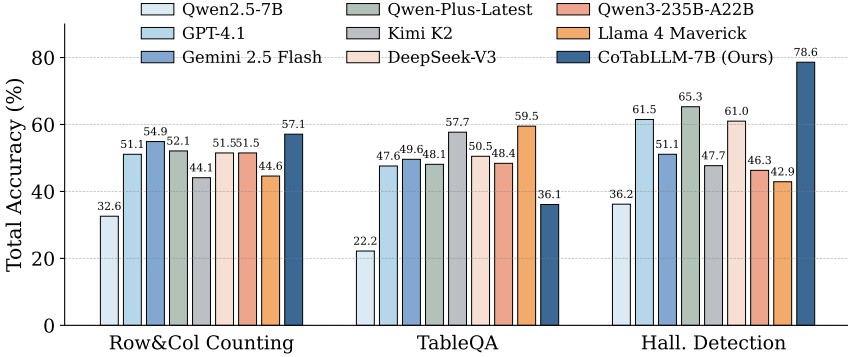

Figure 3: A head-to-head comparison of our fine-tuned model against leading non-thinking LLMs.

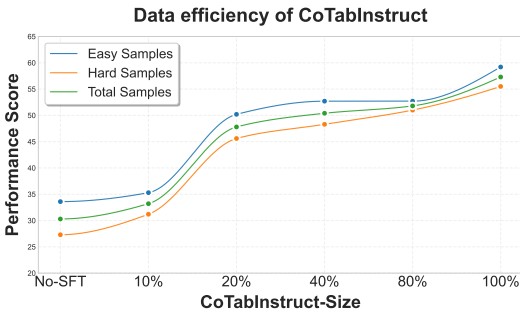

Figure 4: Performance curves of CoTabInstruct under different supervised fine-tuning data scales. Increasing the proportion of training data consistently improves model performance across easy, hard, and total samples.

| Metric | Dataset | LLM | Human | PCC |
|--------|---------|-----|-------|-----|
| DTD | InfoTabs | 2.54 | 2.67 | |
| | WTQ | 2.05 | 2.10 | 0.988 |
| | CoTabBench | 2.60 | 2.87 | |
| CCC | InfoTabs | 2.95 | 3.24 | |
| | WTQ | 2.67 | 2.54 | 0.982 |
| | CoTabBench | 3.12 | 3.42 | |
| SDC | InfoTabs | 3.02 | 3.20 | |
| | WTQ | 3.25 | 3.17 | 0.986 |
| | CoTabBench | 4.47 | 4.53 | |
| **Overall PCC** | | | | **0.973** |

Table 5: Consistency evaluation between LLM-as-a-judge and human expert ratings on 200 random sample subsets. Each metric is reported across three datasets with Pearson correlation coefficients (PCC).

Table 4 provides a comprehensive evaluation of over 20 leading LLMs on CoTabBench. The results immediately reveal a significant performance challenge, with even SOTA models like Qwen-Plus-Latest and GPT-4.1 achieving modest overall scores of 55.2 and 53.4, respectively. Our experiments demonstrate that targeted fine-tuning on CoTabInstruct is more critical than raw model scale for this domain. Notably, our **CoTabLLM-7B** model, with a score of 57.3, outperforms these larger SOTA counterparts, powerfully validating our approach. This stark contrast suggests that the widespread performance drop is not merely a flaw in model architecture, but exposes a fundamental blind spot in the TableQA field: a prevalent focus on simple grids has left models undertrained for the dual challenges of Structural Irregularity and Content Heterogeneity.

**Performance Analysis by Task Type**    A closer look reveals distinct challenges for each task. Even for the seemingly straightforward **RowCol. Counting** task, non-reasoning SOTA models like Qwen-Plus-Latest struggle, achieving a score of 52.1. This indicates a fundamental difficulty in parsing complex table structures. **TableQA** proves to be a major bottleneck for nearly all models, highlighting the difficulty of grounding queries in irregular layouts. Finally, performance on **Hallucination Detection** varies significantly, though it is here that our fine-tuned **CoTabLLM-7B** achieves its highest score of 78.6, suggesting our training method effectively improves factual grounding.

**Impact of Table Complexity on Performance**    The effectiveness of our Complexity Assessment Framework is confirmed by the clear gap between easy and hard subsets: most models degrade on hard tables (e.g., DeepSeek-R1 drops from 72.3 to 66.6), while the gap nearly vanishes on Row and Column Counting because it relies on Structural Irregularity and is less affected by Content Heterogeneity. Complementing this, the data-scaling curves in Figure 4 show that performance consistently improves as the proportion of CoTabInstruct used for SFT increases, demonstrating the dataset's effectiveness in enhancing model performance on complex tables.

## 5    CONCLUSION

In this work, we address the critical gap between current TableQA benchmarks and the complex, weakly-structured tables found in real-world documents. We introduced CoTabBench, a large-scale benchmark designed to systematically evaluate model performance against the dual challenges of Structural Irregularity and Content Heterogeneity. Our extensive experiments reveal that even SOTA LLMs falter on CoTabBench, yet our fine-tuned 7B model, CoTabLLM, achieves comparable performance to leading models after training on our companion dataset, CoTabInstruct. This demonstrates that the primary bottleneck is not model architecture, but a scarcity of representative training data. We release our benchmark and training corpus to the community to spur the development of models capable of robust, real-world table understanding.

## 6    ETHICS STATEMENT

This research is based on datasets collected from publicly available and open-source tables on various websites. All data were carefully curated and reviewed to ensure no personally identifiable information was included, and efforts were made to avoid any data issues. The study adheres to ethical standards regarding data privacy and transparency, with a commitment to minimizing biases and ensuring fairness in the model's development. No conflicts of interest or sponsorship concerns are present.

## 7    REPRODUCIBILITY STATEMENT

To ensure the reproducibility of our work, we will open-source several resources to the community. These include the dataset, CoTabBench, along with the source code, pretrained weights, and training configurations for our model, CoTabLLM. Detailed instructions for reproducing the experiments, including data processing steps, model architecture, and evaluation protocols, will be provided in the supplementary materials. These resources will enable others to replicate our experiments and build upon our work.

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

## A    THE USE OF LLMs

In this work, large language models (LLMs) were only employed as auxiliary tools to support the writing process. Their usage was limited to tasks such as language polishing, sentence restructuring, grammar refinement, consistency checking, and final error correction. At no stage were LLMs used to generate research ideas, experimental designs, or substantive content. Importantly, no fabricated data, references, or scientific claims were introduced through LLM usage. All conceptual contributions, methodological designs, experimental implementations, and analytical results presented in this paper are entirely the responsibility of the authors. The authors take full responsibility for the accuracy and integrity of this work.

## B    RELATED WORK

The field of Table Question Answering (TableQA) has been significantly advanced by a series of foundational benchmarks. Early works such as WTQ (Pasupat & Liang, 2015b; Herzig et al., 2020) and TabFact (Chen et al., 2020) established the groundwork by using Wikipedia-based tables to test models' capabilities in compositional semantic parsing and fact verification. These datasets were instrumental in stimulating initial research; however, they primarily feature well-structured tables where answers can often be directly extracted from table cells, which does not fully represent the complexity of real-world use cases.

Subsequent research has aimed to address these limitations by introducing more sophisticated challenges. One line of work has focused on generating more natural, free-form answers. For instance, FeTaQA (Nan et al., 2022) requires models to synthesize information from multiple, sometimes discontinuous, table cells to generate descriptive answers, pushing beyond simple fact extraction. Another direction has been to incorporate multi-modal reasoning. Datasets like HybridQA (Chen et al., 2021b) and TAT-QA (Zhu et al., 2021) require models to perform multi-hop reasoning over both tabular and unstructured textual data, mirroring how information is often consumed in practice.

Furthermore, a significant body of research has concentrated on domain-specific challenges and complex numerical reasoning. Benchmarks such as FinQA (Chen et al., 2021c) for financial reports, AIT-QA (Katsis et al., 2022) for the airline industry, and TABMWP (Lu et al., 2023) for mathematical problems have highlighted the difficulties models face with specialized vocabularies and multi-step numerical calculations. Concurrently, the structural complexity of tables has also been explored. HiTab (Cheng et al., 2022), for example, specifically targets the challenges posed by hierarchical tables common in statistical reports, which require navigating complex index structures.

However, these benchmarks largely address structural complexity and content heterogeneity in isolation, failing to capture real-world tables that are simultaneously weakly-structured and contain diverse data types. CoTabBench is the first to systematically address this compound challenge. By providing a large-scale benchmark of tables from authentic domains exhibiting both properties, our work establishes a more rigorous and realistic testbed to evaluate the true robustness of modern LLMs.

**Benchmarks for Table-based Reasoning**    The field of Table Question Answering (TableQA) has grown substantially, driven by the development of robust datasets that engage advanced algorithms in semantic comprehension tasks (Huang et al., 2024; Li et al., 2023c; 2024d). Early benchmarks, such as WTQ (Pasupat & Liang, 2015a), SQA (Iyyer et al., 2017), and TabFact (Chen et al., 2020), set the cornerstone for TableQA research. They are primarily based on HTML tables from Wikipedia, which are relatively well-structured. The question-answering pairs in these benchmarks often rely

on extracting factoid-level information from specific cells, which does not fully represent the multi-dimensional queries posed in real-world scenarios.

To bridge this gap, subsequent benchmarks have expanded the task's challenges in several directions. One line of work focuses on generative answering; ToTTo (Parikh et al., 2020), OTTQA (Chen et al., 2021a), and FeTaQA (Nan et al., 2022) introduce free-form question-answer datasets that require models to generate answers beyond the explicit content of the table, better aligning with the open-ended nature of real-world questions. Another direction targets complex reasoning in specific domains. For instance, FinQA (Chen et al., 2021c) and AIT-QA (Katsis et al., 2022) emphasize numerical computation skills on financial tables, requiring models to not only interpret but also to compute information precisely. Furthermore, to evaluate logical reasoning capabilities, benchmarks like WikiSQL (Zhong et al., 2017), Spider (Yu et al., 2018), and Bird (Li et al., 2023a) use logical expressions (e.g., SQL) as supervisory signals. Concurrently, the structural complexity of tables has also been addressed, with benchmarks like HiTab focusing specifically on hierarchical table understanding.

Despite these advances, existing benchmarks tend to address structural complexity and content heterogeneity in isolation. However, real-world tables, particularly those found in scientific papers or financial reports, often exhibit both weakly-structured features and content heterogeneity simultaneously. Our proposed **CoTabBench** is the first benchmark designed to systematically address this compound challenge, incorporating real-world complexities into its evaluation scenarios to effectively address the limitations of existing benchmarks.

**Methods for Table Question Answering**  As a significant research area, the methodology of TableQA has undergone substantial evolution (Mueller et al., 2019; Jin et al., 2022). Early methods were centered on semantic parsing, aiming to convert natural language questions into executable SQL queries (Pasupat & Liang, 2015a; Wang et al., 2021). While effective in certain contexts, these approaches heavily depend on large-scale annotated data and struggle with unstructured content within tables. Subsequently, pre-trained models such as TAPAS (Herzig et al., 2020), TaBERT (Yin et al., 2020), and OmniTab (Jiang et al., 2022) significantly enhanced the deep semantic understanding of tables through large-scale pre-training.

In recent years, the advent of Large Language Models (LLMs) has tremendously advanced the TableQA field (Zhuang et al., 2024; Zha et al., 2023; Su et al., 2024; Zhang et al., 2024; Singha et al., 2023; Li et al., 2023b; Lei et al., 2023; He et al., 2023; Qwen et al., 2025; Bai et al., 2023; Yang et al., 2024; Li et al., 2022; 2024c). The research focus has shifted to leveraging the powerful in-context learning and generalization capabilities of LLMs. To adapt LLMs for this task, researchers have explored various strategies, including fine-tuning (Li et al., 2024b; Su et al., 2024; Zhuang et al., 2024) and training on specialized instruction datasets (Zhang et al., 2024). To handle more complex reasoning chains, the community has proposed several chain-of-thought-style frameworks. For instance, Chain-of-Table (Wang et al., 2024b) explicitly simulates reasoning steps by generating a series of intermediate tables, while ReAcTable (Zhang et al., 2023) integrates external tools like Python or SQL for step-by-step computation. Although these state-of-the-art methods have pushed the upper limits of model reasoning, their design and validation have predominantly been based on relatively well-structured tables. Therefore, their effectiveness when confronted with the highly irregular and content-diverse tables featured in our **CoTabBench** remains a critical, unverified question, highlighting the urgent need for a new benchmark that reflects real-world complexity.

## C  BENCHMARK CONSTRUCTION DETAILS

To ensure the transparency of our work and the reproducibility of our study,we provides a detailed exposition of the CoTabBench construction pipeline.

### C.1  COMPLEX TABLE COLLECTION

To ensure the authenticity and challenge of our benchmark, we collected raw tables from two highly complementary sources, sourcing them in their native formats to preserve their intrinsic complexity. Specifically, we extracted tables from scientific literature as **LaTeX** source and from public websites as **HTML**, thereby capturing the dual challenges of structural complexity and content heterogeneity.

**Scientific Literature**  Our first source comprises tables from academic documents, for which we utilized the **DocGenome** (Xia et al., 2024) dataset. Crucially, we extracted tables directly from their native **LaTeX** source code (i.e., the `\begin{tabular}...\end{tabular}` environment). This approach guarantees the preservation of complex structural information encoded by commands such as `\multicolumn` and `\multirow`—semantics that are often degraded or lost during the conversion to PDF or other formats. Our collection primarily draws from fields on arXiv known for dense, data-rich tables, including Computer Science (cs.CV, cs.AI, cs.LG), Physics (physics.data-an), and Economics (econ.EM).

**Public Websites**  To complement the structured tables from academia, our second source consists of tables gathered from diverse public websites, where data is predominantly presented in **HTML** format. This approach introduces significant heterogeneity in both content and layout, mirroring real-world web data. Using a general web crawler built with Feapder and BeautifulSoup, we targeted data-intensive domains, including: *(i) Government and Statistics* (e.g., official public information catalogs and related disclosures), *(ii) Finance and Business* (e.g., corporate annual reports), and *(iii) Medicine and Health* (e.g., with a focus on information regarding medical drugs, such as from pharmaceutical databases and treatment portals). This ensures our benchmark evaluates models on a wide spectrum of table styles encountered in the wild.

A final, critical step in our collection process was securing a visual representation for every table candidate. For tables sourced from **LaTeX**, we extracted both the table's source text and its corresponding image. Similarly, for **HTML** tables, we obtained a corresponding image by rendering the page in a browser and capturing a screenshot. This dual-modality approach, which including text and image, was essential to validate that the extracted source text corresponds to a well-formed table and to provide an intuitive visual reference that significantly streamlined the subsequent manual filtering process.

Through this dual-source strategy, we ensured that CoTabBench comprehensively covers a spectrum from the rigorous, highly structured tables found in academic research to the more free-form and content-diverse tables encountered on the real-world web.

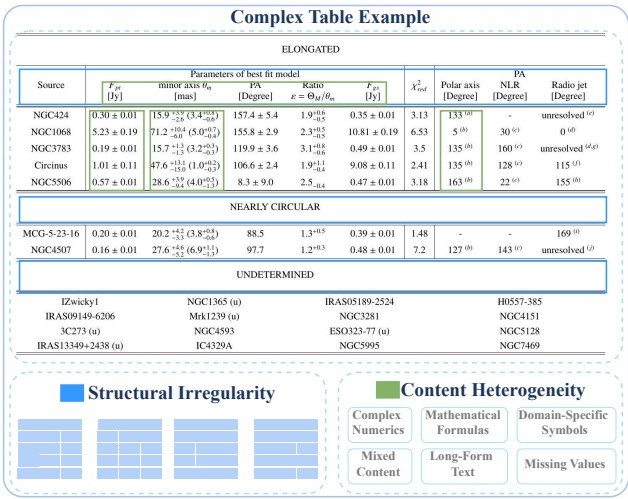

Figure 5: A comprehensive overview of complex table

## C.2 THREE-STAGE FILTERING PROCESS

The collected raw table corpus contained significant noise, such as non-data tables used for web layout or incorrectly parsed tables. To address this, we designed and implemented a rigorous three-stage filtering process.

**Stage 1: Rule-based Filtering**  This initial stage applied a set of deterministic heuristics to rapidly discard a large volume of clearly unsuitable candidates. The rules included: (i) **Size Filtering**,

which removed tables with fewer than 3 rows or 2 columns; (ii) **Content Density Filtering**, which discarded tables with an average of fewer than 2 characters per cell or an empty cell ratio exceeding 80%; and (iii) **Trivial Content Filtering**, which eliminated tables where all cells contained identical content.

**Stage 2: Model-based Filtering**   Surviving candidates were then subjected to a dual-model assessment using both a Large Language Model (LLM) and a Multimodal Large Language Model (MLLM). The process was twofold. First, both models performed a critical **validity check** to filter out non-data tables. The LLM assessed the markdown text to identify and discard tables used purely for page layout or navigation, while the MLLM examined the rendered image to reject visually malformed structures or elements incorrectly parsed as tables. Second, tables that passed this validity screen underwent a **quality assessment**. The LLM evaluated the table's content for semantic coherence and informational value, while the MLLM validated the visual integrity, checking for issues like severe cell misalignment or overlapping text that would render the table incomprehensible.

**Stage 3: Human Verification**   As the final quality gate, all tables that passed the automated filtering stages were visually inspected by trained human annotators. Following a detailed guideline, annotators performed a definitive review to identify and remove any remaining low-quality or subtly flawed samples that the models may have missed. This step was crucial for guaranteeing the final quality and utility of the benchmark.

## D   TASK DESIGN AND DEFINITION

The tasks for CoTabBench were designed to rigorously evaluate a model's ability to overcome the intrinsic challenges of weakly-structured tables. While the tasks necessitate multi-hop reasoning to interpret complex layouts, they deliberately de-emphasize complex numerical computation. This approach isolates the core challenge of table understanding, shifting the locus of difficulty from arithmetic to the essential capabilities of structural parsing and information extraction from irregular layouts.

| | | FB15K237 + *pagelink* | | | | | |
| | | node2vec | | | LINE | | |
| | BEM | KG | BG | concat | KG | BG | concat |
|---|---|---|---|---|---|---|---|
| TransE | O | 85.59 | 75.12 | 89.39 | 85.59 | 77.57 | 89.44 |
| | I | 85.51 | 82.56 | 85.97 | 86.35 | 85.44 | 87.05 |
| | P | **88.89** | **86.32** | **90.29** | **88.21** | **86.27** | **90.01** |
| TransD | O | 86.06 | 75.12 | 89.18 | 86.06 | 77.57 | 89.00 |
| | I | 83.73 | 78.86 | 84.16 | 86.58 | 85.10 | 86.69 |
| | P | **88.60** | **85.39** | **89.90** | **88.70** | **85.30** | **89.73** |

| | | FB15K237 + *desc* | | | | | |
| | | doc2vec | | | sentence2vec | | |
| | BEM | KG | BG | concat | KG | BG | concat |
|---|---|---|---|---|---|---|---|
| TransE | O | 85.32 | 75.62 | **87.92** | 85.32 | 83.42 | 88.43 |
| | I | 86.19 | 81.50 | 86.41 | 87.61 | 85.18 | 88.07 |
| | P | **87.68** | **81.52** | 87.86 | **88.05** | **85.82** | **88.57** |
| TransD | O | 85.83 | 75.62 | 88.07 | 85.83 | 83.42 | 88.52 |
| | I | 86.75 | 81.44 | 86.85 | 87.96 | 84.97 | 88.07 |
| | P | **87.34** | **82.24** | **88.15** | **88.36** | **86.12** | **88.86** |

Figure 6: An example of a structurally complex table from our benchmark. Its physical dimensions (18x8) differ significantly from its logical structure, which is the target of Task 1.

**Task 1: Row and Column Counting**   This fundamental task tests a model's comprehension of a table's **logical structure** versus its mere physical layout. In the presence of merged cells, a simple count of physical rows or columns is often misleading. Models are required to parse the headers to identify the true number of logical entities, directly probing their ability to deconstruct a table's visual representation and grasp its underlying schema. As illustrated in Image 6, the table exhibits a logical row count of 18 and a logical column count of 8.

**Task 2: TableQA** This task evaluates **Table Question Answering (TableQA)** over weakly structured, heterogeneous tables. A system must read a table, ground the query to the correct evidence, and generate the final answer through light reasoning. Concretely, models are required to (i) parse multi-level headers and merged cells—e.g., cells defined via \multirow and \multicolumn—to recover the full row–column context of target entries; (ii) perform schema-aware localization to identify all relevant cells, including those referenced by hierarchical or implicit headers; and (iii) execute minimal yet essential operations, such as exact lookup, comparison, selection, and simple aggregation (e.g., max/min, sum, ratio), while handling units, ranges, footnotes, and special symbols. Because table layouts often deviate from a canonical grid, correct answers cannot be obtained by naive coordinate indexing. Instead, models must combine structural understanding with content normalization and multi-hop grounding across header hierarchies to produce faithful, table-supported answers.

**Task 3: Hallucination Detection** This task assesses a model's factual fidelity by presenting two types of negative questions, probing its ability to resist making false inferences induced by misleading layouts or plausible-sounding queries.

- Data Absence: The model is queried for information that is explicitly not present in the table. A correct response must affirm the absence of the information.
- Attribution Error: The model is presented with a value that exists in the table but is incorrectly attributed to the wrong row or column header. A correct response must identify this attribution error, demonstrating precise factual grounding.

# E   MULTI-AGENT QA ANNOTATION FRAMEWORK

To address the dual challenges of prohibitive manual annotation costs and the qualitative shortcomings of single-model generation, we propose a Multi-Agent QA Annotation Framework. This framework orchestrates a cascaded pipeline of specialized agents designed to systematically generate, filter, and validate high-fidelity QA pairs. The workflow begins with generation and bifurcates based on an internal consistency check, with all paths ultimately converging at a final, rigorous validation stage. This architecture ensures both scalability and reliability for complex data annotation tasks.

**Question and Initial Answer Generation** The process commences with a **Question Generator** agent. This agent ingests tabular data and is intentionally directed via specialized prompts to focus on information-dense and structurally complex regions, such as multi-span cells. For each question it formulates, the agent also generates a provisional *First Answer* based on its initial interpretation of the source data.

**Question Filtering** Subsequently, the generated question undergoes a stringent evaluation by a **Question Filter** agent. This stage serves as a crucial quality gate, systematically discarding invalid queries. It eliminates questions that are improperly formatted, deviate from predefined task requirements, or conflate multiple distinct inquiries into a single query. Only well-formed and task-compliant questions proceed.

**Information Extraction** For each valid question that passes the filtering stage, an **Extractor** agent is invoked. Its function is to parse the question and retrieve all relevant content from the source table. This step ensures that any subsequent answer generation is strictly grounded in the provided tabular information, effectively mitigating the risk of model-internal hallucination.

**Refined Answer Generation** Using the extracted information as the sole source of context, an **Answer Generator** agent formulates a *Second Answer*. By restricting the generator's knowledge base to this curated, relevant information, this step aims to produce a more precise and factually constrained answer to the valid question.

**Answer Consistency Verification** A critical self-verification step, **Answer Consistency** checking, is then performed. This module programmatically compares the *First Answer* (from the Question Generator) with the *Second Answer* (from the Answer Generator). If the two answers are

determined to be semantically consistent, the QA pair is provisionally approved and advanced directly to the final validation stage. If a discrepancy is detected, the pair is flagged and escalated for discrepancy resolution.

**Multi-LLM Voting for Discrepancies**    QA pairs that fail the consistency verification are routed to the **Multi-LLMs Voting** module. Here, the question is posed to a panel of three independent LLMs, which generate their answers. The answer that receives the majority of votes is selected as the final candidate answer. This ensemble method provides a robust mechanism for resolving ambiguities and correcting errors from the initial generation stages.

**Human Check**    To guarantee the highest level of accuracy for the most challenging cases, answers determined by the Multi-LLMs Voting module are subsequently presented for **Human Check**. A human annotator reviews the question, the voted answer, and the source table to provide final verification. This step is indispensable for ensuring the correctness of nuanced or complex questions where automated consensus may still be insufficient.

**Multi-Dimensional Automated Validator**    As the terminal gatekeeper of the framework, the **Validator** module performs a final, exhaustive verification on every QA pair, regardless of whether it passed via Answer Consistency or the Multi-LLM Voting and Human Check pathway. This validation is conducted across three critical, orthogonal dimensions:

- **Data Consistency:** This dimension ensures absolute data grounding by programmatically verifying that all factual assertions posited in the final answer have a direct, verbatim correspondence within the source table. The validator tokenizes the answer into constituent claims and executes a substring matching algorithm against the entirety of the table's content. A successful validation requires that every piece of factual information, be it a name, a category, or a specific value, can be traced back to an identical string in the source data. This stringent check acts as a definitive safeguard against data fabrication and model hallucination, ensuring that the answer introduces no extra-tabular information.

- **Hierarchical Consistency:** To prevent misattribution errors, this check verifies the referential integrity between the question's specification and the answer's origin. It performs a rigorous bidirectional mapping. First, it parses the question to identify explicit or implicit coordinate constraints (e.g., specific row headers, column headers, or their intersection). It then verifies that the content of the answer is located precisely at these coordinates within the table. Concurrently, it performs the reverse check: it locates the answer's content within the table and verifies that its corresponding coordinates align with the entities specified in the question. This dual-directional approach confirms that the answer is not only factually correct but is also correctly attributed to the subject of the query.

- **Numerical Consistency:** For any question entailing quantitative reasoning (e.g., summation, averaging, counting, or comparison), this dimension guarantees computational integrity. The validator first parses the question to determine the required mathematical operation and the operand values, which it then extracts from the table. Subsequently, it dynamically generates a transient Python execution script. This script performs the identified calculation on the extracted numerical data and embeds an `assert` statement that compares the computed result against the numerical value present in the model's answer. The execution of this script serves as a formal proof of the answer's numerical accuracy.

## F    COMPLEXITY ASSESSMENT FRAMEWORK

To empirically validate the assertion that CoTabBench presents a more formidable challenge than existing benchmarks, we move beyond anecdotal evidence and introduce a novel, quantitative framework for assessing table complexity. Existing benchmarks, while progressively incorporating more difficult reasoning tasks, are built upon tables that remain structurally simple and content-wise homogeneous. This discrepancy obscures the primary hurdles encountered in real-world scenarios. Our framework is designed to systematically deconstruct and measure the intrinsic difficulty of a table along two fundamental, orthogonal dimensions: **Structural Irregularity** and **Content Heterogeneity**. This not only provides a principled basis for comparing CoTabBench against its predecessors but also offers a standardized methodology for future research on complex table understanding.

**Quantifying Structural Irregularity via a Hybrid Approach** The quantification of a table's structural irregularity is operationalized through a hybrid framework that synergistically combines deterministic parsing with model-based semantic analysis. For metrics amenable to direct pattern matching, such as the Merged Cell Ratio (MCR) and Merged Cell Span (MCS), we employ a rule-based engine. This engine parses the table's source code (e.g., HTML or LaTex) and utilizes complex regular expressions to precisely identify and count cells with `rowspan` and `colspan` attributes, providing an objective and highly efficient measure of basic layout deviations. Conversely, for assessing the Header Depth Index (HDI), which demands a deeper comprehension of the table's hierarchical organization, we leverage an LLM. We instruct the LLM, via a specialized prompt, to function as an expert in table structure analysis. Its task is to parse the table's layout and directly infer the maximum hierarchical depth of both the row and column headers. The sum of these two inferred depth values constitutes the Header Depth Index (HDI), enabling a robust quantification of hierarchical complexity that circumvents the limitations of rule-based systems.

**Quantifying Content Heterogeneity via LLM-as-a-Judge** In contrast to structure, quantifying Content Heterogeneity requires assessing nuanced semantic properties, a task for which simple rules are ill-suited. We therefore adopt an LLM-as-a-judge methodology, leveraging the advanced evaluative capabilities of a powerful large language model (*e.g.*, Qwen-Plus-Latest) to serve as a consistent evaluator. The process is governed by a meticulously engineered prompt that instructs the model to score a serialized representation of the table along three predefined axes: Cell Content Complexity (CCC), Data Type Diversity (DTD), and Semantic & Domain Complexity (SDC). The prompt furnishes the LLM with a detailed rubric for each axis, including explicit definitions and a 1-to-5 scoring scale with clear exemplars. By structuring the task in this manner, we elicit a standardized JSON output containing the three scores, thereby transforming a qualitative assessment into a reproducible, quantitative vector that represents the table's content complexity.

## G EXPERIMENT SETUP AND REPRODUCIBILITY DETAILS

**Models Details** To ensure fair and reproducible evaluations across all models, we implemented a standardized experimental protocol. For each question in the CoTabBench benchmark, we used a uniform prompt template that instructed the model to return its answer in a structured JSON format. We enforced this output structure using JSON Schema where the model's API supported it. The final answer was then programmatically parsed from the model's generated JSON object. Proprietary models (e.g., GPT-4.1, Qwen-Plus-Latest) and major open-source models were queried via their official APIs, using the specific model versions detailed in Table 6. Specialized table-finetuned models, including our own, were deployed locally on a server equipped with four 4x4090 GPUs. Inference was managed by the vLLM framework for efficiency and speed.

**Fine-tuning Details** Our model, CoTabLLM, was developed by applying Supervised Fine-Tuning (SFT) to the Qwen2.5-7B-Instruct model. We utilized the Low-Rank Adaptation (LoRA) technique to efficiently adapt the model to the task. The fine-tuning process was conducted on our CoTabInstruct dataset, which contains over 11,000 complex tables and 32,000 question-answer pairs. The training was configured with a batch size of 16 and a maximum sequence length of 4096 tokens to accommodate the large context of the tables and questions. All training was performed on the same 8xL20 GPU server setup.

| Model Name | Model Version |
|---|---|
| Qwen-Turbo-Latest | qwen-plus-2025-04-28 |
| Qwen-Plus-Latest | qwen-turbo-2025-04-28 |
| DeepSeek-R1 | deepseek-r1-0528 |
| DeepSeek-V3 | deepseek-v3-0324 |
| GPT-4.1 | gpt-4.1-2025-04-14 |
| GPT-4.1-mini | gpt-4.1-mini-2025-04-14 |
| o4-mini | o4-mini-2025-04-16 |
| gemini-2.5-flash | stable version |

Table 6: List of models and their specific version used in the evaluation.

