# OpenReview forum: "CoTabBench: A Real-World Benchmark for Question Answering over Weakly-Structured and Heterogeneous Tables"
_ICLR.cc/2026/Conference — Submitted to ICLR 2026_

### Official Review · Reviewer_5E9K · 2025-10-19

**Soundness:** 2
**Presentation:** 2
**Contribution:** 3
**Rating:** 6
**Confidence:** 4

**Summary:**

This paper introduces CoTabBench, a large-scale benchmark designed to evaluate LLMs on complex, real-world table question answering. Addressing the gap left by existing benchmarks that use well-structured data, CoTabBench features over 2,700 weakly-structured and heterogeneous tables from 10 domains. Experiments reveal a significant performance drop in state-of-the-art LLMs on this benchmark. However, the authors' fine-tuned 7B model, CoTabLLM, trained on their new 11,000-table CoTabInstruct dataset, outperforms even top models like GPT-4.1. This suggests the primary bottleneck for real-world table understanding is the lack of representative training data, not model architecture.

**Strengths:**

1. The authors introduce CoTabBench, a new large-scale, multi-modal table question-answering dataset. They also propose quantitative metrics to measure and validate its structural irregularity and content heterogeneity.
2. The authors constructed an instruction-tuning dataset, CoTabInstruct, which is shown to effectively improve model performance.
3. Extensive experiments are conducted to demonstrate the challenging nature of the proposed CoTabBench dataset.

**Weaknesses:**

1. Insufficient Comparison with Prior Work: While Tables 1 and 3 offer comparisons to some existing table QA datasets, several prior works have also introduced datasets focusing on structural irregularities and domain-specific knowledge, such as MMTBench [1], SPIQA [2], RealHiTBench [3], and ENTRANT [4]. The manuscript would be strengthened by a clearer discussion of CoTabBench's unique advantages over these datasets. Specifically, the tables in WTQ and InfoTabs, used for comparison in Table 3, are highly structured and sourced from general-domain Wikipedia content. A more compelling comparison against other recent datasets that are claimed to be structurally irregular, real-world, or domain-specific is expected.
2. Lack of In-depth Error Analysis: The experiments lack a detailed analysis of the models' failure cases. For instance, what are the most common error types in the TableQA task? Are they primarily due to difficulties in understanding hierarchical headers, handling multi-row cells, or interpreting domain-specific terminology? I suggest that the authors provide and categorize representative error cases to offer deeper insights.

[1] MMTBENCH: A Unified Benchmark for Complex Multimodal Table Reasoning

[2] SPIQA: A Dataset for Multimodal Question Answering on Scientific Papers

[3] RealHiTBench: A Comprehensive Realistic Hierarchical Table Benchmark for Evaluating LLM-Based Table Analysis

[4] ENTRANT: A Large Financial Dataset for Table Understanding

**Questions:**

1. Table 1, Figure 2, and Figure 3 are not referenced in the main text. Please ensure all tables and figures are properly cited.
2. CoTabBench is a multi-modal dataset, yet the comparisons in Tables 1 and 3 are exclusively with text-only table QA datasets. It is recommended to include comparisons with other multi-modal table QA datasets, such as MMTBench [1], SPIQA [2], and ComTQA [3].
3. The content presented in Figure 3 and Table 4 appears to be redundant. Please consider merging or removing one of them.

[1] MMTBENCH: A Unified Benchmark for Complex Multimodal Table Reasoning

[2] SPIQA: A Dataset for Multimodal Question Answering on Scientific Papers

[3] TabPedia: Towards Comprehensive Visual Table Understanding with Concept Synergy

---

### Official Review · Reviewer_Z8Fj · 2025-10-29

**Soundness:** 2
**Presentation:** 2
**Contribution:** 2
**Rating:** 4
**Confidence:** 4

**Summary:**

The paper introduces CoTabBench, a critically needed, large-scale benchmark for Table Question Answering (TableQA) designed to address the significant disparity between existing datasets, which rely on overly clean, well-structured tables, and the chaotic reality of real-world data. CoTabBench uniquely focuses on the joint challenge of Structural Irregularity and Content Heterogeneity, compiling over 2,700 weakly-structured tables and more than 8,600 question-answer pairs spanning 10 distinct domains, sourced from complex documents such as native LaTeX academic papers and diverse public web domains. The methodology is supported by a novel Complexity Assessment Framework that quantitatively validates the benchmark's rigor, showing superior metrics in both structural and content complexity compared to predecessors like WTQ and InfoTabs. Crucially, extensive experiments involving over 20 state-of-the-art Large Language Models (LLMs) reveal a significant performance degradation, with even highly sophisticated proprietary models like GPT-4.1 achieving a modest overall score of 53.4%. Conversely, the purpose-built CoTabLLM-7B model, trained on the companion CoTabInstruct corpus, surpasses these larger competitors, establishing that the primary constraint in achieving robust, real-world TableQA performance is the scarcity of appropriate instruction data tailored to these compound complexities.

**Strengths:**

1. Bridging the Real-World Data Gap via Compound Complexity CoTabBench uniquely addresses a critical gap by sourcing tables from native LaTeX academic papers and diverse public websites, simultaneously ensuring both structural irregularity (e.g., merged cells) and deep content heterogeneity (e.g., formulas, long text). This dual-source methodology creates a highly authentic and rigorous testbed for real-world Table Question Answering (TableQA) challenges.


2. The Rigorous Multi-Dimensional Complexity Assessment Framework The paper proposes a novel Complexity Assessment Framework that systematically quantifies difficulty along the distinct dimensions of Structural Irregularity and Content Heterogeneity. This framework uses objective metrics like Merged Cell Ratio (98.34%) and Header Depth Index (3.91) to empirically validate CoTabBench’s superior rigor compared to prior datasets.


3. Validation Through Specialized Instruction Tuning and Robust Baselines The creation of the dedicated CoTabInstruct training corpus and the resulting CoTabLLM-7B model definitively proves that the current performance bottleneck is data scarcity, not model scale. CoTabLLM-7B establishes a robust baseline by achieving a higher overall accuracy (57.3%) that surpasses larger proprietary models like GPT-4.1 (53.4%).

**Weaknesses:**

W1: Task Design Limitation: Decoupling Structural Interpretation from Complex Reasoning The task design intentionally simplifies reasoning chains (requiring only "minimal yet essential operations") to isolate structural challenges, which risks making the benchmark an optimized test of structural parsing rather than a holistic measure of real-world TableQA capabilities. This limits the rigorous stress-testing of models' ability to handle complex content like mathematical formulas or synthesizing long-form text, despite the benchmark including these heterogeneous elements.


W2: Reliance on Proprietary LLM-as-a-Judge for Content Quantification A core part of the Complexity Assessment Framework uses the proprietary, closed-source Qwen-Plus-Latest model to quantify Content Heterogeneity (CCC, DTD, SDC). This dependency introduces significant reproducibility risk and transparency limitations, as the specific, nuanced scoring prompts and internal mechanism of the commercial API cannot be independently verified or debugged by the research community.


W3: Insufficient Transparency in Complex Data Sourcing and Filtering The tables collected from "public websites" via a "large-scale crawler" are described vaguely, omitting crucial technical details about the crawler’s methodology, raw data volume, or explicit steps taken to mitigate selection or domain bias across the 10 targeted domains. This lack of transparency limits the ability to replicate the data collection process or fully assess the potential structural or semantic biases inherited in the web-scraped subset.

W4: Ambiguity in Ethical Compliance and PII Handling The Ethics Statement provides only generic assurance that "no personally identifiable information was included" without detailing the specific PII removal protocol. This vagueness is insufficient given the sensitive nature of the real-world source material from domains like corporate annual reports (Finance) and pharmaceutical databases (Medicine), posing potential legal or distribution risks.


W5: Generalization and Overfitting Concerns for CoTabLLM Despite its strong overall score (57.3%), the CoTabLLM-7B model shows a severe performance drop on the core TableQA task (36.1%), suggesting it may be highly specialized to the specific structural and annotation patterns of the CoTabInstruct training corpus. This disparity raises concerns that the model might struggle to generalize its complex reasoning capabilities to novel table structures or reasoning tasks outside of the distribution it was fine-tuned on.


W6: Unverified Generalizability of Complexity Framework Metrics While the Complexity Assessment Framework is validated against only two prior benchmarks (WTQ, InfoTabs), the universal applicability of its calculated metrics (e.g., Merged Cell Span, Header Depth Index) is unverified against benchmarks specifically targeting hierarchical tables (like HiTab) or intense domain-specific reasoning (like FinQA). The framework needs broader testing to prove its long-term utility as a standardized measure for all types of complex tables.

**Questions:**

same as weakness

---

### Official Review · Reviewer_WZdo · 2025-10-31

**Soundness:** 2
**Presentation:** 3
**Contribution:** 2
**Rating:** 4
**Confidence:** 4

**Summary:**

The authors introduce CoTabBench, a new large-scale, multi-domain QA benchmark over weakly‐structured, heterogeneous tables. CoTabBench includes over 2.7K real-world tables and 8.6K+ question–answer pairs spanning 10 domains. The paper proposes a novel complexity‐assessment framework to quantify each table’s structural irregularity and content heterogeneity. To support training, they release CoTabInstruct (~11K tables) and train a 7B model (CoTabLLM) on it; this model even outperforms GPT-4.1 on the CoTabBench tasks. Extensive experiments show that state-of-the-art LLMs suffer marked performance drops on CoTabBench, underscoring the benchmark’s value for advancing robust real-world table understanding.

**Strengths:**

1. CoTabBench fills a clear gap by focusing on “weakly-structured and heterogeneous” tables from real sources. Unlike prior datasets (e.g. WikiTables, TableFact, FinQA) that use clean grids, CoTabBench tables often have merged cells, nested headers, multiline cells, etc., and cover 10 diverse domains (scientific and applied).

2. The paper is well-structured and mostly well-written, with sections organized logically (construction pipeline, tasks, complexity, experiments, etc.)

3. The paper evaluates over 20 models, including state-of-the-art proprietary models (GPT-4.1, Qwen-Turbo, Gemini 2.5) and many open-source LLMs (Llama3/4, Qwen2.5/3, DeepSeek, etc.). Both “non-thinking” and chain-of-thought (“thinking”) modes are tested. This breadth demonstrates the benchmark’s difficulty.

**Weaknesses:**

1. The claim that CoTabLLM-7B “outperforms GPT-4.1” and other proprietory LLMs may be misleading. CoTabLLM is fine-tuned specifically on CoTabInstruct, whereas GPT-4.1 is evaluated off-the-shelf. It’s expected that a model trained on similar data will have an advantage. There should be a comparison or discussion in the paper around state-of-the-art trainable baselines (https://arxiv.org/abs/2402.01155, https://arxiv.org/abs/2107.07653) and few-shot table QA frameworks (https://arxiv.org/abs/2301.13808).

2. The authors introduce CoTabInstruct (11k tables, 32k QAs) but give few details on how it was assembled or how it differs from CoTabBench. For example, it is unclear whether CoTabInstruct tables overlap with the CoTabBench test set or if they were drawn from separate sources/timeframes. If there is any overlap, it could inflate CoTabLLM’s performance. Similarly, how were the 32k QA generated (presumably a similar multi-agent pipeline)? More transparency about the train/validation/test split and annotation process for CoTabInstruct would improve reproducibility.

3. The question-answer pairs are generated in an automated fashion. Therefore, the reliability on the correctness of the answers remains skeptical unless a manual qualitative evaluation is done on a random sample of QA pairs corresponding to the tables.

**Questions:**

Look at the weaknesses above. I have a few more questions enumerated below.

1. Could the authors clarify the source and curation of the CoTabInstruct training set? Specifically, are its tables and QA pairs strictly disjoint from CoTabBench (to prevent leakage)?

2. For better understanding, it would be useful to see concrete examples of each task. Could the authors provide sample QA pairs for (a) row/column counting question, (b) TableQA question (with its reasoning steps), and (c) hallucination question (both Data-Absent and Attribution-Error types)? The descriptions in Sec.2.3 are clear, but actual examples would illustrate the difficulty and clarify the answer format.

3. For the negative (hallucination) questions, how are the two types balanced? Are there equal numbers of Data-Absence vs. Attribution-Error questions? Also, is the model simply asked to answer these questions (where presumably the “correct” answer is some null/negation), or is it a classification (Yes/No) task?

4. Is the code for the data creation pipeline be released? That can help the community create larger scale datasets and adapt the framework to particular use cases of Table QA.

---

### Official Review · Reviewer_TtPK · 2025-11-05

**Soundness:** 1
**Presentation:** 3
**Contribution:** 1
**Rating:** 2
**Confidence:** 4

**Summary:**

This paper presents CoTabBenc, a benchmark for tabular question answering. The authors argue that the existing benchmarks do not capture the complexity of real-world tables (complex structures, noise, etc.) and thus curate a collection of tables from various sources (academic documents, web-pages) and create CoTabBench -- a detaset or 2700+ tables, and 8600+ associated question-answer pairs. They also present a training corpus and present CoTabLLM (fine-tuned Quen on the dataset) and show that it outperforms other LLMs (not fine-tuned on the dataset) on CoTabBench.

**Strengths:**

- The paper is written well and is easy to understand

- The authors evaluate multiple LLMs on the proposed benchmark and present a detailed compartive anlaysis

**Weaknesses:**

The biggest weakness of the work is the lack of details about the benchmark creation process. The paper provides a high level overview of the process, but offers little to no details. No one can read the paper and repeate the becnchmark creation process.

For instance, how the papers from arXiv and websites were chosen for sampling tables?

The benchmark is essentially created following a set of rules and question-answer pairs generated synthetically by an LLM. How does the process ensure quality and diversity of these questions? Further, the questions do not capture real-world nuances -- a key limitation of existing works by the authors.

No details are provided about the human validation process. What is the background of these validators? How de we judge the quality of their work? Are there any inter-annotator studies that are performed?

While the authors have compared performance of multiple LLMs on the proposed benchmark, it would have been better to test different SoTA table question answering methods on the benchmark.

Likewise, authors compare the performance of fine-tuned CoTabLLM with with different LLMs in a zero-shot setting. This is unfair. A model especially trained for a specific task is expected to outperform models not trained for the task.

Also, I could not find a link to the resources developed for the work.

**Questions:**

Please see above.

**Details Of Ethics Concerns:**

Since tables are crawled from web, and sufficient details are not provided, it needs to be checked for potential copytight issues, and if crawling guidelines (such as in robots.txt) were followed.

---

### Meta-Review · Area_Chair_Ju3H · 2025-12-31

**Summary:**

The paper presents a benchmark for tabular question answering, which consists of 2.7K tables and 8.6K QA pairs from 10 domains. Tables are extracted from arXiv and websites. The paper also presents an 11K table training dataset, CoTabInstruct, and use this to fine-tune a 7B model which then outperforms models such as GPT-4.1.

Strengths: the paper's focus on weakly-structured, heterogenous tables from real sources is well-motivated. The evaluation of a large number of models, both closed and open, and thinking/non-thinking, is great.

But, there are major concerns (see below).

**Reviewer Concerns:**

One major concern (from reviewers TtPK and WZdo) is the lack of detail on the construction of the datasets: both the eval dataset and the training dataset. Some crucial missing details are whether there is overlap between the eval (CoTabBench) and training sets (CoTabInstruct) which could inflate performance, and how the papers from arXiv and websites were chosen for sampling tables.

The QA pairs are also synthetically LLM-generated, and there is missing detail about the human validation of the dataset.

The authors did not respond to these concerns and questions, which also gives me major reservations about the work.

**Reviewer Scores:**

There was no author response, so I expect that the scores would remain 2 / 4 / 4 / 6.

---

### Decision · Program_Chairs · 2026-01-26

Reject